# Latent Linear ODEs with Neural Kalman Filtering for Irregular Time Series Forecasting

## Abstract

Over the past four years, models based on Neural Ordinary Differential Equations have become state of the art in the forecasting of irregularly sampled time series. Describing the data-generating process as a dynamical system in continuous time allows predictions at arbitrary time points. However, the numerical integration of Neural ODEs typically comes with a high computational burden or may even fail completely. We propose a novel Neural ODE model that embeds the observations into a latent space with dynamics governed by a linear ODE. Consequently, we do not require any specialized numerical integrator but only an implementation of the matrix exponential readily available in many numerical linear algebra libraries. We also introduce a novel state update component inspired by the classical Kalman filter, which, to our knowledge, makes our model the first Neural ODE variant to explicitly satisfy a specific self-consistency property. It allows forecasting irregularly sampled time series with missing values and comes with some numerical stability guarantees. We evaluate the performance on medical and climate benchmark datasets, where the model outperforms the state of the art by margins up to 30%.

## 1 Introduction

Continuous dynamical systems described by ordinary differential equations (ODE) propagate a given state into any time in the future. Hence ODE based models are natural candidates for the task of forecasting irregularly sampled time series. Furthermore many real world systems are well described by ODEs. Since the seminal paper by Chen et al. (2018) Neural ODEs have become building blocks of state of the art models for irregularly sampled time series forecasting. To predict a future state an ODE model would need an estimation of the present state and then propagate the state by solving an initial value problem. The present work proposes a model that introduces novel ideas both with respect to the state estimation and to the propagation. One serious issue with Neural ODEs is the cost and possible failure of the numerical integration. There exist many numerical schemes for this purpose, but in any case the cost of the integration for a required accuracy depends on the analytical properties of the right hand side and can become arbitrarily large or lead to failure. This is a serious problem for Neural ODEs, which has been tackled by different types of regularizations (Finlay et al., 2020; Ghosh et al., 2020; Kelly et al., 2020).

We propose a model where the observations are nonlinearly mapped into a latent space and a linear ODE with constant coefficients describes the latent dynamics. Solving the initial value problem simplifies to taking the matrix exponential, for which efficient and stable numerical implementations are available. According to Koopman operator theory (Brunton et al., 2022) such linear ODEs are expressive enough to approximate nonlinear ODEs. Furthermore, such linear dynamics are well understood and can be analyzed and modified using tools from linear algebra.

For the state estimation we propose a filter inspired by the classical Kalman filter that updates the state given a new observation. However, it does not operate in the linear latent domain, but in the observation domain, and it is not probabilistic. The filter is designed to deal in a natural way with missing values and satisfies a self-consistency condition, such that the model state will only change at an observation if it differs from the model prediction.

To the best of our knowledge our model is the first model that gives provable guarantees of forward stability at intialization.

We evaluate the model on three benchmark datasets for forecasting irregularly sampled time series with missing values (USHCN, MIMIC-III, MIMIC-IV) and improve on the existing models by a considerable margin in all cases.

The contributions of this work are as follows:

(1) We provide a joint view of many ODE-based and related models as latent state space models with four different model components system, filter, encoder and decoder, which by design can handle irregular time series data.

(2) We formulate and argue five different desiderata for the properties of such models, esp. having fast and simple system components / ODE integrators, self-consistency, forward stability and handling missing values.

(3) We propose a model consisting of a linear ODE system and a Kalman-like filter, LinODEnet, and show that it guarantees to fulfil these desidered properties.

(4) In experiments on forecasting three different irregular time series datasets with missing values we show that LinODEnet reduces the error by up to 30% over the previous state of the art.

## 2  RELATED WORK

In the last decades a variety of different forecasting models with principally different architectures such as recurrent neural networks (RNNs), attention based models, convolution based models, variational autoencoders and generative adversarial networks and others have emerged (see appendix D and Benidis et al. (2020) for a recent survey). Most closely related to our work are models based on ODEs, models using Kalman Filtering, Koopman theory and Neural Flows.

**ODE based models.** A neural ODE is a model that is specified as the solution to an ordinary differential equation of the form

$$\frac{\mathrm{d}}{\mathrm{d}t}x(t) = f(t, x(t), \theta)$$

where $f$ is a neural network parametrized by $\theta$. Neural ODEs were pioneered by Chen et al. (2018), who derived a continuous time analogue of the backpropagation algorithm that allows for effective training of such models. The GRU-D model (Che et al., 2018) could be considered an even earlier version of a neural ODE since it coincides with linear differential equation of the form $\frac{\mathrm{d}}{\mathrm{d}t}x(t) = \mathrm{diag}(-\lambda_1, \ldots, -\lambda_m)x(t)$.

Since then there has been tremendous development with a large amount of offspring models. ODE-RNNs and Latent Ordinary Differential Equations (Latent-ODE) were introduced by Rubanova et al. (2019), combining the dynamics of an ODE with an RNN to perform a state space update at the point of a new observation. GRU-ODE-Bayes (De Brouwer et al., 2019) is an improvement over the standard ODE-RNN that parametrizes the ODE component as a continuous time analogue of a GRU, increasing stability. Neural Controlled Differential Equations (NCDE; Kidger et al. (2020)) use a spline smoothing technique in order to create a continuous reconstruction of the time series before feeding it to the ODE. This model however was only used for classification tasks. Neural ODE Processes (NDPs; Norcliffe et al. (2020)) extend the framework of Neural Processes (NPs; Garnelo et al. (2018b;a)) to the class of Neural ODE models. Similarly, Neural Jump Stochastic Differential Equations (NJSDE; Jia & Benson (2019)) combine Neural ODEs with Stochastic Processes. Finally, the recent Linear State-Space Layers (LSSL; Gu et al. (2021)) interpret both RNNs and convolutions as special cases of linear state space layers, and construct deep LSSLs by stacking multiple such layers.

**Kalman Filtering.** The Kalman Filter (Kalman, 1960) provides optimal incremental state updates for linear differential equations with Gaussian noise. While the original version requires complete observations, a modified Kalman filter copes with missing values (Cipra & Romera, 1997), which is one of the motivations for our filter design.

The Normalizing Kalman Filter (de Bézenac et al., 2020) is closest in spirit to our model, it proposes a discrete time linear Gaussian model and an invertible observation map given by a normalizing flow. As ours the model can cope with missing values, but it is not apt for irregularly sampled time series.

KalmanNet (Revach et al., 2022) is another model for regularly sampled time series that uses an RNN to calculate the Kalman gain. It is meant to present a robust alternative to other approaches of nonlinear Kalman filtering (extended Kalman filter) and not a general times series forecasting algorithm. Some other work on Kalman filtering by neural networks (Wilson & Finkel, 2009; Millidge et al., 2021) either addresses very special cases or are related to the modelling of the brain and rather far away from our work.

**Koopman Theory.** A Koopman Operator (Koopman, 1931) is a linear operator on a space of time dependent functions that describes the propagation of observations of a dynamical system through time. While these function spaces are infinite dimensional, in many cases there exist useful finite dimensional Koopman operator approximations that can be created by various methods, that have been summarized in a recent review (Brunton et al., 2022). Such Koopman representations have been combined with Kalman filters for the linear operator (Netto & Mili, 2018), but the linear representation is obtained by a classical method which does not work for irregularly sampled observations with missing values. Some works use neural networks to learn Koopman representations (cf. section 5.4 of Brunton et al. (2022)), but have largely different architectures and are not applied to irregularly sampled time series with missing values. The model we present can be seen as learning an approximate Koopman operator, but it is outside the scope of this work to exploit this representation along the lines of Koopman theory.

**Neural Flows.** Neural Flows (Biloš et al., 2021) are another related model class. The authors propose replacing ODEs by invertible time dependent diffeomorphisms and use different parametrizations for such diffeomorphisms (ResNet, GRU, coupling flow). This is similar to and in special cases would amount to learning the solution of a differential equation instead of learning the right hand side. They mention also the parametrization of the solutions by a matrix exponential, but then opted for other parametrizations.

## 3 PROBLEM FORMULATION

A *time series dataset $D$* is a set of *time series instances*, sequences $(t_i, x_{t_i}^{\text{obs}}) \in (\mathbb{R} \cup \mathcal{O})^*$ encoding observations $x_{t_i}^{\text{obs}}$ at time $t_i$. The observation space $\mathcal{O}$ usually is just composed of $M$ channels: $\mathcal{O} := \mathbb{R}^M$. If observations in some channels can be missing, we write $\mathcal{O} := (\mathbb{R} \cup \{\text{NaN}\})^M$ and dinstinguish it from the space $\mathcal{X} := \mathbb{R}^M$ of complete observations.

The *time series forecasting problem* is, given a time series dataset $D$ from an unknown distribution $q$, a loss $\ell : \mathcal{O}^* \times \mathcal{X}^* \to \mathbb{R}$ on observations, and a function split : $\mathbb{R}^* \to (\mathbb{R}^* \times \mathbb{R}^*)^*$ that splits the time points of a time series into (possibly multiple) pairs of two subsequences, pasts $p$ and futures $s$, to find a model $\hat{x} : \mathbb{R}^* \times (\mathbb{R} \times \mathcal{O})^* \to \mathcal{X}^*$ that for given future time points and past observations, predicts future observations, minimizing the expected loss

$$\mathbf{E}_{(t,x^{\text{obs}}) \sim q} \left[ \frac{1}{|\text{split}(t)|} \sum_{(p,s) \in \text{split}(t)} \ell\big(x_s^{\text{obs}}, \hat{x}(s, (p, x_p^{\text{obs}}))\big) \right]$$

A simple index-based split function just outputs all possible splits into a past of $P$ time points and $F$ future time points, and the loss usually is just an instance-wise loss, e.g., the instance-wise mean squared error:

$$\text{split}^{\text{index}}(t; P, F) := \Big( \big((t_{i-P+1}, \ldots, t_i), (t_{i+1}, \ldots, t_{i+F})\big) \mid i \in P{:}|t| - F \Big), \quad P, F \in \mathbb{N}$$

$$\ell^{\text{MSE}}(x^{\text{obs}}, \hat{x}) := \frac{1}{|x^{\text{obs}}|} \sum_{i=1}^{|x^{\text{obs}}|} ||\text{diag}(\text{not-missing}(x_i^{\text{obs}}))(x_i^{\text{obs}} - \hat{x}_i)||_2^2$$

where not-missing$(x_i^{\text{obs}})$ yields the indicator vector for $x_{i,m}^{\text{obs}} \neq \text{NaN}$.

## 4 LATENT STATE SPACE MODELS

First, consider a general class of latent state space models that follow the schema from Algorithm 1. That is, the model has a state estimate $\hat{x}_t$ which is encoded into a latent state $\hat{z}_t$ and propagated

forward in time by a system component. For each observation, latent state is decoded and updated with a filter component.

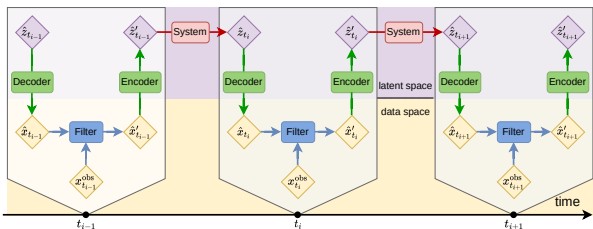

Figure 1: Latent State Space Model

**Algorithm 1** Latent State Space Model.

> **Input:** Query times $(s_j)_{j=1\ldots m}$,
> Observations $D = (t_i, x_{t_i}^{\text{obs}})_{i=1:n}$
> **Parameters:** Initial latent state $c$.
>
> $\widetilde{D} \leftarrow \text{sort}\left(D \cup \{(s_j, \texttt{NaN})\}_{j=1:m}\right)$
> $t_0 \leftarrow t_1;\ z_{t_0} \leftarrow c$
> **for** $t_i, x_{t_i}^{\text{obs}}$ in $\widetilde{D}$ **do**
>     $\Delta t_i \leftarrow t_t - t_{i-1}$
>     $\hat{z}_{t_i} \leftarrow \textbf{\textcolor{red}{System}}(\Delta t_i,\ z'_{t_{i-1}})$
>     $\hat{x}_{t_i} \leftarrow \textbf{\textcolor{green}{Decoder}}(\hat{z}_{t_i})$
>     $\hat{x}'_{t_i} \leftarrow \textbf{\textcolor{blue}{Filter}}(\hat{x}_{t_i},\ x_{t_i}^{\text{obs}})$
>     $\hat{z}'_{t_i} \leftarrow \textbf{\textcolor{green}{Encoder}}(\hat{x}'_{t_i})$
> **end for**
> **Return:** Estimated states $(\hat{x}'_{t_i})_{i=1:n}$.

This setup is indeed a very general model class, Table 1 shows how many current state space models can be described by this schema.

Neural ODEs, introduced in 2018 by Chen et al., use a ordinary differential equation to represent the system component. The vector field which describes the right hand side of the ODE is given by a neural network $f$:

$$\dot{z} = f(t, z(t)), \qquad z(t_0) = z_0 \implies z(t) = \text{odeint}(f, z_0, [t_0, t]) \tag{1}$$

Crucially, they introduced a continuous version of backpropagation through time that allows to compute gradients for a Neural ODE model with respect to a loss function by solving a so called adjoint equation backwards in time. This allows one to compute gradients without implementing a differentiable ODE-integrator and without back-propagating through the integrator steps.

Table 1: Comparison of Continuous Time Latent State Space Models

| Model | System Component | Filter Component |
|---|---|---|
| GRU-D | $\dot{z}(t) = -\text{diag}(\max(0, w_i)) \odot z(t)$ | $x'_t \leftarrow m_t x_t^{\text{obs}} + (1 - m_t)x_t$ 
 $z'_t \leftarrow \text{GRUCell}(x'_t, z_t)$ |
| ODE-RNN | $\dot{z}(t) = \text{NN}(t, z(t))$ | $z'_t \leftarrow \text{RNNCell}(x_t^{\text{obs}}, z_t)$ |
| GRU-ODE-Bayes | $\dot{z}(t) = \text{GRUCell}(0, z(t)) - z(t)$ | $z'_t \leftarrow \text{GRUCell}(f(x_t^{\text{obs}}, m_t, z_t), z_t)$ |
| NCDE | $\dot{z}(t) = g(z(t))\dot{s}(t)$ | $s(t) \leftarrow \text{CubicSpline}(t \mid (t_i, x_i)_{i=1\ldots n})$ |
| Neural Flow | $F(\Delta t, x) = x + \varphi(\Delta t)g(\Delta t, x)$ | Bayesian Filtering |
| LinODEnet (ours) | $\dot{z}(t) = Az(t)$ | $x'_t = \text{KalmanCell}(x_t^{\text{obs}}, x_t)$ 
 $\quad = x_t - f(m_t \odot (x_t - x_t^{\text{obs}}))$ |

By design, latent state space models can handle irregularly sampled time series. We formulate five further principled desiderata and argue for them in the following:

- D1. *fast (and simple) integrators*: solving the ODE system for propagating the latent state should be fast (and simple).
- D2. *self-consistency*: the model does not change its forecast when observing its own predictions.
- D3. *forward stability*: at initialization the model maps data with zero mean and unit variance to outputs with zero mean and unit variance for arbitrarily long sequences.
- D4. *can handle missing values*: the model can handle missing values in the observations.

D5. *observed channels can influence all other channels*: observed values in one channel can influence the estimation of unobserved states of all channels.

**D1. Fast (and simple) integrators.** Neural ODE models only faithfully represent the solution to an ODE when the numerical integrators use sufficiently small adaptive step-sizes (Ott et al., 2020). Thus, fitting generic Neural ODEs is a challenging task since during training the ODE can become stiff, which forces adaptive step-size integrators to take minuscule time steps. Towards this end, several remedies are available: Ghosh et al. (2020) and Finlay et al. (2020) propose temporal regularization terms, and many models implicitly address this issue by choosing a special form of vector field. An overview is in Table 1. For example, GRU-ODE-Bayes uses a GRUCell with a tanh activation function that induces global Lipschitz continuity, which increases stability.

**D2. Self consistency.** All of the proposed models use different filter mechanisms to update the state estimate when new observations are recorded. However, in our analysis we noted that a classical Kalman Filter (Kalman, 1960) satisfies a *self-consistency* property that none of the published Neural ODE models seems to incorporate.

We say a point-forecasting model is *self-consistent*, if and only if the model does not change its forecast when observing its own predictions. More specifically, given a forecasting model $\hat{y}(t \mid D)$, where $D = \{(t_i, x_i) \mid i = 1 \ldots n\}$ is the set of observations, then $\hat{y}$ is self-consistent if and only if for all finite sets of generated predictions $\hat{D} = \{(s_j, \hat{y}(s_j)) \mid j = 1 \ldots m\}$

$$\hat{y}(t \mid D) = \hat{y}(t \mid D \cup \hat{D}) \tag{2}$$

For a probabilistic models $p(y(t) \mid D)$ we similarly define self-consistency as the condition

$$\mathbf{E}_{\hat{y} \sim p(y(t)|D)}[\hat{y}] = \mathbf{E}_{\hat{y} \sim p(y(t)|D \cup \hat{D})}[\hat{y}] \tag{3}$$

Note that in this case it is expected that $\mathrm{Var}_{\hat{y} \sim p(y(t)|D \cup \hat{D})}[\hat{y}]$ should decrease with the size of $\hat{D}$.

**D3. Model stability.** Model stability is crucial in order to allow training over long sequences without issues of divergence.

**Definition 1** (forward stability). We say a function $f \colon \mathbb{R}^n \to \mathbb{R}^m$ is *forward stable*, if and only if it maps data with zero mean and unit variance to outputs with zero mean and unit variance.

$$\forall i : \begin{bmatrix} \mathbf{E}_{x \sim p}[x_i] \\ \mathbf{V}_{x \sim p}[x_i] \end{bmatrix} = \begin{bmatrix} 0 \\ 1 \end{bmatrix} \implies \forall j : \begin{bmatrix} \mathbf{E}_{x \sim p}[f(x)_j] \\ \mathbf{V}_{x \sim p}[f(x)_j] \end{bmatrix} = \begin{bmatrix} 0 \\ 1 \end{bmatrix} \tag{4}$$

Similarly, one can define *backward stability* as the condition that the gradients, or more precisely the vector-Jacobian product maps data with zero mean and unit variance to gradients with zero mean and unit variance. Typically, the random distributions of network parameters are chosen in a way to ensure either forward- or backward stability at initialization or a compromise between the two at as in general simultaneous forward and backward stability is impossible He et al. (2015). For example, Attention layers (Vaswani et al., 2017) introduce a scaling factor of $1/\sqrt{d}$ and Dropout (Srivastava et al., 2014) multiplies the input by the reciprocal of the droprate.

However, recently a new approach has emerged that achieves both in a ResNet architecture, by simply introducing an additional single scalar parameter initialized with 0 that masks the non-linearity, making the model look like an identity map. We will refer to this as the *ReZero*-technique (Bachlechner et al., 2021), although previous works also showed similar ideas Hayou et al. (2021); Skorski et al. (2021); De & Smith (2020); Zhang et al. (2018); Balduzzi et al. (2017); Shang et al. (2017)

$$\text{ReZero:} \qquad x \leftarrow x + \alpha f(x) \qquad \alpha: \text{learnable scalar initialized with } 0 \tag{5}$$

In particular recent research suggests that this techniques allows one to refrain from using batch-normalization layers (Ioffe & Szegedy, 2015). We use variants of the ReZero-technique throughout all model components.

**D4. Can handle missing values.** The model can handle missing values. For latent state space models the filter needs to be able to do so.

**D5. Observed channels can influence all other channels.** This is crucial as often channels are correlated with each other, hence observing one channel can provide information about all other channels.

# 5 Latent Linear ODEs with Neural Kalman Filtering (LinODEnet)

We propose two specific innovations for ODE-based latent state space models: (i) to use a linear ODE for the system component and (ii) to use a Kalman like filter component, such that the overall model fulfills the desiderata D1 to D5, and call it LinODEnet. LinODEnet is structured as shown in Algorithm 1. We describe its components in turn.

## 5.1 System Component

To avoid having to use complicated numerical integrators, LinODEnet uses a simple homogeneous linear ODE with constant coefficients. This has the huge advantage that the solution can be expressed in closed form in terms of the *matrix exponential*.

**Definition 2** (Linear ODE). If the vector field is an affine function of the state vector the ODE is called *linear*, i.e. if and only if it is of the form:

$$\dot{x}(t) = A(t) \cdot x(t) + b(t) \qquad x(t_0) = x_0 \tag{6}$$

for some matrix valued function $A \colon \mathbb{R} \to \mathbb{R}^{n \times n}$ and vector-valued function $b \colon \mathbb{R} \to \mathbb{R}^n$. If $A$ and $b$ are constant, we call it a linear ODE with *constant coefficients*. If $b = 0$, we say it is *homogeneous*.

**Lemma 1** (Solution of Linear ODE). *The solution of a homogeneous linear ordinary differential with constant coefficients can be expressed in term of the matrix exponential*

$$\dot{x}(t) = Ax(t) \iff x(t + \Delta t) = e^{A \Delta t} x(t) \tag{7}$$

*Proof.* See for instance Teschl (2012). □

In particular, implementations of the matrix exponential are readily available in many popular numerical libraries such as SciPy (Virtanen et al., 2020), TensorFlow (Abadi et al., 2016) or PyTorch (Paszke et al., 2019). Typical implementations such as scaling and squaring approaches by Higham (2005) and Al-Mohy & Higham (2009) offer high performance and tight error bounds, establishing desideratum D1.

A second advantage of the linear system is the possibility to parametrize or regularize the kernel matrix in order to achieve certain properties. We highlight the initialization with a skew-symmetric matrix as of particular importance, and we use it as the default initialization in all experiments.

**Lemma 2.** *If $K$ is skew-symmetric, then $e^K$ is orthogonal and $h(t) = e^{K \cdot t} x$ is forward stable for all $t$ and $x \sim \mathcal{N}(\mathbf{0}_n, \mathbb{I}_n)$.*

Motivated by these properties, we define the *LinODECell* (Alg 2). Note that crucially, in comparison to the GRU-D model, a general latent linear ODE model allows for imaginary eigenvalues of the system matrix, corresponding to oscillatory system behaviour. In the GRU-D model, the authors intentionally restricted the model to a non-positive, real diagonal matrix.

---

**Algorithm 2** LinODECell

**Input:** scalar time delta $\Delta t$
latent state $z_t$
**Parameters:** matrix $K$
zero-initialized scalar $\varepsilon$
parametrization $\psi$

$z_{t+\Delta t} \leftarrow e^{\varepsilon \cdot \psi(K) \cdot \Delta t} \cdot z_t$
**Return:** latent state $z_{t+\Delta t}$

---

## 5.2 Filter Component

Any state space model must have a way of incorporating new measurements as they come in.

**Definition 3.** We call a function of the form $f \colon \mathcal{O} \times \mathcal{X} \to \mathcal{X}, \ (x^{\text{obs}}, \hat{x}) \mapsto \hat{x}'$ a *filter*. If the observation space $\mathcal{O}$ contains NaN values we say it *allows for missing observations*. If the state space $\mathcal{X}$ is equal to the non-missing part of the observation space $\mathcal{O}$, we call it *autoregressive*. Finally, we say a filter *cross-correlates channels*, if even the observation of just a single channel can potentially update the state estimate in all channels.

One of the big achievements of classical filtering theory is the Kalman Filter (Kalman, 1960), which is the provably optimal state update in terms of squared error loss when the system consisting of normally distributed variables evolving according to a linear dynamical system.

Assuming the state is distributed as $x \sim \mathcal{N}(\mu_t, \Sigma_t)$, at time $t$, and evolves according to a linear dynamical system $\dot{x}_t = A_t x_t + w_t$, then since the family of Normal distributions are closed under linear transformations, the state is normally distributed for all times $t$. Given a noisy measurement $y_t = H_t x_t + v_t$ with $R_t = \mathbf{E}[v_t v_t^\top]$, which is only partially observed according to a mask $m_t$, then the optimal state update is (Cipra & Romera, 1997)

$$\mu_t' = \mu_t - \Sigma_t H_t \Pi_t (H_t \Sigma_t H_t^\top + R_t)^{-1} \Pi_t (H_t \mu_t - \tilde{y}_t) \tag{8a}$$

$$\Sigma_t' = \Sigma_t - \Sigma_t H_t \Pi_t (H_t \Sigma_t H_t^\top + R_t)^{-1} \Pi_t H_t \Sigma_t \tag{8b}$$

Where $\Pi_t = \mathrm{diag}(m_t)$, and $\tilde{y}_t$ is $y_t$ where the missing values were replaced with arbitrary values. Inspired by this formula, we introduce the *linear and non-linear KalmanCell* which can be used as drop-in replacements for regular RNN-, GRU- or LSTMCells.

$$\text{linear KalmanCell:} \qquad \hat{x}_t' \leftarrow \hat{x}_t - \alpha \widetilde{B} H^\top \Pi_t \widetilde{A} \Pi_t (H \hat{x}_t - x_t^{\mathrm{obs}}) \tag{9a}$$

$$\text{non-linear KalmanCell:} \qquad \hat{x}_t' \leftarrow \hat{x}_t - \varepsilon \phi (B H^\top \Pi_t A \Pi_t (H \hat{x}_t - x_t^{\mathrm{obs}})) \tag{9b}$$

In both cases $A, B, H$ are learnable weight matrices. In the linear case we introduce a special parametrizations $\widetilde{A} = \mathbb{I} + \varepsilon_A A$ and $\widetilde{B} = \mathbb{I} + \varepsilon_B B$. Here $\varepsilon, \varepsilon_A, \varepsilon_B$ are learnable scalars, that are initialized with zero, which ensures forward stability. $\phi$ is an arbitrary neural network with $\phi(0) = 0$. By design the Kalman-Cell can handle NaN values, for implementation details see 4 and 5, establishing D4.

**Lemma 3** (KalmanCell at initialization). *At initialization, the non-linear KalmanCell is the identity function. The linear KalmanCell's behaviour is dependent on the choice of $\alpha$: if $\alpha = 1$, it always*

---

**Algorithm 3** Non-Linear KalmanCell

**Input:** Current State estimate $\hat{x}_t \in \mathbb{R}^n$, observed datapoint $x_t^{\mathrm{obs}} \in \mathbb{R}^m$
**Parameters:** Learnable matrices $A, B, H$ zero-initialized scalar $\varepsilon$, neural network $\phi$.
**Options:** If autoregressive, $m = n$ and $H = \mathbb{I}_n$.

$\Pi_t \leftarrow \mathrm{diag}(\text{not-missing}(x_t^{\mathrm{obs}}))$
$\hat{x}_t' \leftarrow \hat{x}_t - \varepsilon \phi(B H^\top \Pi_t A \Pi_t (H \hat{x}_t - x_t^{\mathrm{obs}}))$
**Return:** Updated state estimate $\hat{x}_t'$.

---

*updates the state to the last observed value, whereas if $\alpha = 0$ is carries the first observed value through. The choice $\alpha = \frac{1}{2}$ corresponds to the classical Kalman Filter (cf. Appendix B.2)*

Note that (9) is different from both Szirtes et al. (2005), Wilson & Finkel (2009), and the recent pre-prints Millidge et al. (2021) and Revach et al. (2022). Moreover, since the KalmanCells do, in contrast to the GRU-D model, use full and not diagonal matrices, D5 is satisfied.

Moreover it is possible to stack multiple filters $(f_i)_{i=1\dots k}$.

$$\text{Stacked Filter:} \qquad \hat{x}_t^{(i+1)} = f_i(x_t^{\mathrm{obs}}, \hat{x}_t^{(i)}) \quad \text{for i=1\dots k} \tag{10}$$

With regards to desideratum D2, there is a strong relationship to the setup of the filter component.

**Definition 4.** We say an autoregressive filter $F \colon \widetilde{\mathcal{X}} \times \mathcal{X} \to \mathcal{X}, (x^{\mathrm{obs}}, \hat{x}) \mapsto \hat{x}'$ is *idempotent*, if and only if it returns the original state estimate as-is whenever all non-missing observations agree with it.

$$x_i^{\mathrm{obs}} = \hat{x}_i \forall i : x_i^{\mathrm{obs}} \neq \texttt{NaN} \implies F(x^{\mathrm{obs}}, \hat{x}) = \hat{x} \tag{11}$$

**Lemma 4.** *If a latent state space model (Alg. 1) is self-consistent, then its filter must be idempotent.*

*Proof.* It this wasn't the case, then $\hat{y}(t \mid D) \neq \hat{y}(t \mid D \cup \{t, \hat{y}(t \mid D)\})$. $\qquad\square$

## 5.3 Overall Model Properties

**Proposition 1.** *If in Algorithm 1 the system component represents a dynamical system, and the filter component is idempotent, and the encoder is left-inverse to the decoder, then the model is self-consistent.*

**Corollary 1.** *LinODEnet is self-consistent at initialization. If the encoder is left-inverse to the decoder it is self-consistent throughout training, establishing desideratum D3.*

The proofs can be found in Appendix B. Table 2 summaries the properties of git adLinODEnet in comparison to other models.

Table 2: Comparison of foreasting model features (†: transformer model)

| model | TFT[†] | NKF | GRU-D | GRU-ODE-Bayes | Neural Flow | LinODEnet |
|---|---|---|---|---|---|---|
| missing-values | ✓ | ✓ | ✓ | ✓ | ✓ | ✓ |
| continuous time | ✓ | ✗ | ✓ | ✓ | ✓ | ✓ |
| global existence | \<NA\> | ✓ | ✓ | ✗ | ✓ | ✓ |
| self-consistency | ✗ | ✗ | ✗ | ✗ | ✗ | ✓ |
| forward stability | ✗ | ✗ | ✗ | ✗ | ✗ | ✓ |
| coupled channels | \<NA\> | ✗ | ✗ | ✓ | ✓ | ✓ |

*Remark* 1 (LinODEnet with hidden state). Since LinODEnet can parse `NaN` values, we consider a small modification consisting of concatenating a number of dummy channels, completely filled with `NaN`-values to all input data. This allows the model to have a working memory in case that the state space does not capture the full dynamics.

## 6 EMPIRICAL EVALUATION

### 6.1 EXPERIMENTS ON IRREGULAR TIME SERIES

While there are many publications dealing with irregular time series for classifications or imputation tasks, there are few that approach with irregular sampled time series forecasting natively. We identify GRU-ODE-Bayes (De Brouwer et al., 2019) and Neural Flow (Biloš et al., 2021).

Irregular time series occur naturally whenever collections of sensor devices sample data independently, sometimes as vastly different rates. This is for example the case with clinical records such as the MIMIC-III (Johnson et al., 2016) and MIMIC-IV (Johnson et al., 2021) datasets or in weather observations such as the USHCN dataset (Menne et al., 2015). We use the same data-preprocessing and evaluation protocol as our baselines De Brouwer et al. (2019) and Biloš et al. (2021). The task is to predict the next 3 samples using an observation horizon of 36 hours.

Table 3: Average MSE and standard deviation across 5 cross-validation folds.
†: results reported by De Brouwer et al. (2019) ‡: results reported by Biloš et al. (2021)

| Model | USHCN | MIMIC-III | MIMIC-IV |
|---|---|---|---|
| NeuralODE-VAE[†] | $0.96 \pm 0.11$ | $0.89 \pm 0.01$ | – |
| NeuralODE-VAE-Mask[†] | $0.83 \pm 0.10$ | $0.89 \pm 0.01$ | – |
| Sequential VAE[†] | $0.83 \pm 0.07$ | $0.92 \pm 0.09$ | – |
| GRU-Simple[†] | $0.75 \pm 0.12$ | $0.82 \pm 0.05$ | – |
| GRU-D[†] | $0.53 \pm 0.06$ | $0.79 \pm 0.06$ | – |
| T-LSTM[†] | $0.59 \pm 0.11$ | $0.62 \pm 0.05$ | – |
| GRU-ODE-Bayes[†] | $\underline{0.43 \pm 0.07}$ | $\underline{0.48 \pm 0.01}$ | – |
| Neural Flow (GRU)[‡] | – | $\underline{0.49 \pm 0.004}$ | $\underline{0.364 \pm 0.008}$ |
| LinODEnet w/ hidden | $\mathbf{0.29 \pm 0.06}$ (↑33%) | $\mathbf{0.45 \pm 0.02}$ (↑5%) | $\mathbf{0.274 \pm 0.002}$ (↑25%) |
| LinODEnet w/o hidden | $\mathbf{0.31 \pm 0.05}$ (↑31%) | $\mathbf{0.45 \pm 0.02}$ (↑5%) | $\mathbf{0.274 \pm 0.002}$ (↑25%) |

Table 3 shows that LinODEnet outperforms all baselines by significant margin. We also observe that in the USHCN dataset, adding hidden channels (Remark 1) gives an additional lift. We suspect that this is due to the numbers of channels being very small (5) for this dataset.

**Training Details.** We use the ADAMW-optimizer (Loshchilov & Hutter, 2018), a variant of the popular ADAM-optimizer Kingma & Ba (2015) that provides a correction when using weight decay. For the filter we use a stack of a linear KalmanCell and two nonlinear KalmanCells. The full hyperparameter selection is in Appendix E.2.

**Reproducibility.** We created two pip-installable python packages: `tsdm` provides utilities for dataset acquisition, pre-processing and a library of encoders. A reference implementation of the model in PYTORCH is available as the package `linodenet`. The experimental code is in a separate repository.

## 6.2 OBSERVATIONS

During training, we consistently observed the emergence of correlation between the rows and columns of the system-components kernel-matrix (Figure 4). This indicates that the matrix get close to a low-rank matrix. Since the rank itself cannot be computed in a numerically stable manner, we considered a smoothed relaxation known as the *effective rank* (Roy & Vetterli, 2007).

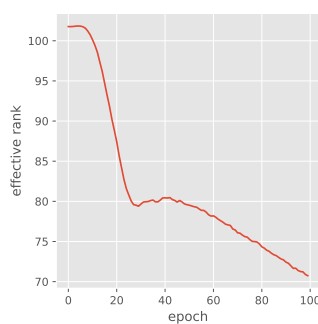

**Definition 5.** The *effective rank* of a matrix is defined as the exponential of the entropy $\mathrm{erank}(A) = e^{H(p)}$, where $p = \sigma / \|\sigma\|_1$ is the discrete probability distribution given by normalizing the singular values $\sigma$ of $A$.

Figure 2

Figure 3 shows the evolution of the spectrum of the kernel matrix for a sample run on the USHCN dataset. One can see that the eigenvalues stay close to being purely imaginary. We speculate that this is due to the main dynamics are essentially periodic in nature, as weather patterns repeat over time

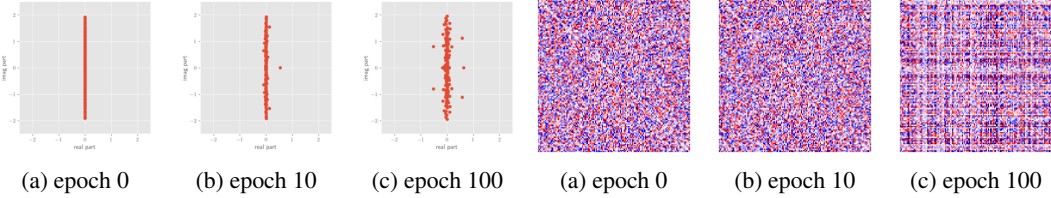

(a) epoch 0     (b) epoch 10     (c) epoch 100          (a) epoch 0     (b) epoch 10     (c) epoch 100

Figure 3: Evolution of the kernel spectrum.          Figure 4: Evolution of the kernel values.

## 7 CONCLUSIONS

We propose a novel forecasting model for irregularly sampled time series with missing values, that maps the observation space to latent space with constant linear ODE dynamics and performs state estimations by an update rule inspired by the Kalman filter. For solving the linear ODE we do not need numerical integration but just matrix exponentials, for which stable and efficient implementations exist. Forward stability of the model at intialization is guaranteed. The model is evaluated and most of the existing forecasting benchmarks for irregularly sampled time series and improves on existing models by a considerable margin. The model opens a way for interesting future work: It naturally allows for future covariates in the forecasting problem. The linear representation of the dynamics allows modification and analysis by means of linear algebra.

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
