# OpenReview forum: "Latent Linear ODEs with Neural Kalman Filtering for Irregular Time Series Forecasting"
_ICLR.cc/2023/Conference — Submitted to ICLR 2023_

### Official Review · Reviewer_kNhy · 2022-10-18

**Confidence:** 4
**Correctness:** 3
**Technical Novelty And Significance:** 2
**Empirical Novelty And Significance:** 2
**Recommendation:** 5

**Clarity, Quality, Novelty And Reproducibility:**

__Clarity + Quality:__
- There is no mention in the body about the "global existence" property highlighted in table 2.
- Some of the notations used in this paper are non-standard and confusing, at least to me:
   - What does $\psi(K)$ mean in Algorithm 2? Is it meant to be $K(\psi)$? i.e. matrix a $K$ parameterised by $\psi$.
   - The star $^*$ notation (for example, $\mathbb{R}^*$) used in section 3 is confusing. Please explain what this means.
- Please clarify which function is required to have model stability (definition 1). Is it just required for the filter component?
- Notation for the filter is inconsistent. For example, in definition 3, a lower case $f$ is used, whereas in Definition 4, an upper case $F$ is used. Note that the vector field of an ODE is also denoted $f$ in this paper, further adding to the confusion.
- The labels in Figure 3 are too small. Please make it bigger.
- What is the `kernel matrix' mentioned in §6.2?
- Some comments on the Appendix:
   - In the proof of self-consistency (section B), please include a line about why the (stacked) filter is idempotent.
   - Section B.1 appears to be incomplete.
   - I am not sure if the argument in section B.2 makes sense. In particular, why do you take $R=\Sigma$? This is generally not true in Kalman filters. I would say the claim "the choice $\alpha=1/2$ corresponds to the Kalman filter" in Lemma 3 is misleading, as this would imply that the two are equivalent in general, although the proof seems to suggest that the correspondence only holds for a very special case of KF.

__Novelty:__
I believe that the ideas introduced in this work are original, in particular the use of linear dynamics in the latent and the Kalman-style update cells, although the technical aspects are not new.

__Reproducibility:__
Sufficient information on the experiments is provided in the appendix for reproducibility, along with dataset and code.


**Strength And Weaknesses:**

__Strengths:__
- The proposed model seems to preform quite well on the benchmark datasets, outperforming the state-of-the art in the climate example by a large margin. Though this may simply be due to the higher dimensionality of the latent space used here compared to that in the baselines (perhaps the baselines may perform just as well if they used hidden states of similar size?), the experiments at least demonstrate that the linear dynamics assumption in the latent space is not so limiting.

- The use of a Kalman filter-inspired update rule is nice, enabling some level of interpretability (at least in the linear, shallow case) as opposed to the RNN-type updates used in the other works. The theoretical benefits gained through this choice is also an advantage.

__Weaknesses:__
- Uncertainty quantification does not seem possible within the proposed framework, which may be useful in several applications. Other models such as ODE-VAE and GRU-ODE-Bayes has an advantage in this regard.

- The experiments section feels a little rushed and lacking in quality. For example,
   - One of the reported benefits of LinODEnet is the computational efficiency due to the use of matrix exponentials. A comparison of compute time between the different models would have been desirable to verify this claim. This claimed efficiency is not so clear since there much be some tradeoff between the efficiency gain by linearity vs the need to lift the dynamics to a higher dimension.
   - A novelty of this paper is the introduction of a neural Kalman filter, which distinguishes itself from RNN-style updates. An ablation study to show the efficacy of a self-consistent filter would have been desirable, e.g. by using the same latent dynamical model and encoder/decoder but with different filters.
   - Likewise, some experiments showcasing the performance of the learned filter would have been a great addition, to demonstrate how useful it can be in online settings to perform long-term predictions. For example, including a plot of the filtered trajectory or showing the improvements in e.g. MSE after applying the update step will suffice.

- The writing in some places is not very clear and the presentation can be improved (more details below).


**Summary Of The Paper:**

In this paper, the authors propose a latent ODE model for irregularly sampled timeseries forecasting, where the dynamics in the latent space evolve according to a linear neural ODE. This is enabled by (1) a nonlinear encoder/decoder pair that maps the dynamics between the latent and observed space, and (2) a neural Kalman-like filter operating in the observation space, which updates the states given an observation. By taking the latent space to be higher dimensional than the observation space, the authors justify the use of a linear ODE, reflecting ideas in Koopman theory. The resulting model satisfies a list of desiderata given by the authors and performs well on some benchmark datasets.

**Summary Of The Review:**

The paper presents a model that, while technically not quite advanced, is original and performs well on several benchmark tasks, proving its capability for use in time series modelling. The paper however suffers from several presentation issues and a weak experiments section.
In light of this, I recommend a light reject, with the possibility of an acceptance provided some of these issues are addressed.

---

> ### Author Response · Authors · 2022-11-19
> **Response to Review by kNhy**
>
> **Regarding uncertainty quantification.** Uncertainty quantification is possible via the use of Normalizing Flows, and we are actively working on a second paper that introduces this as well as additional features such as explicit treatment of covariates and static metadata.
>
> **Regarding computational efficiency.** The review is right that there is a tradeoff between the encoding and the simpler latent dynamics. As we mentioned, one main benefit is that there are very good algorithms for the matrix exponential available, whereas for generic ODEs it is unclear which numerical integrator to use. If one looks at how the matrix exponential algorithms work (cf. “Computing the Action of the Matrix Exponential” by Al-Mohy and Higham) then the internal step-size depends on the largest eigenvalue of A, which can easily be controlled through regularization, whereas regularizing general ODEs is complicated (cf. “How to Train Your Neural ODE” by Finlay et al.).
>
> Our model can be trained within a day on all benchmark datasets using consumer hardware (RTX 3090).
>
> **Regarding Filter component.** We added ablation studies for using a linear filter and a non-idempotent filter.
>
> **Regarding experimental showcase.** We added a forecast plot with filtered trajectories.
>
> **Regarding forward stability.** All model components need to be forward stable at initialization in order for the full model to be forward stable at initialization. For the encoder, we use a ReZero-ResNet (cf. “ReZero is All You Need” by Bachlechner et al.), for the system matrix one can either use a skew-symmetric initialization, or introduce a trainable ReZero scalar. We do both such that it is easy to substitute different initialization schemes for the matrix. For the filter, we use a ResNet-like architecture that also makes use of ReZero scalars.
>
> **Regarding notation.** $Ψ$ refers to an optional kernel parametrization, for example, skew-symmetric parametrization would be $ψ(K) = ½(K-K^⊤)$, such that the system component would be $(∆t，x) ↦ exp(ε½(K-K^⊤)∆t)x$ instead of  $(∆t，x) ↦ \exp(εK∆t)x$. This ensures that the system matrix has only imaginary eigenvalues, i.e. the dynamics of the system are periodic in the latent space. We added an ablation experiment using diagonal parametrization to highlight differences to GRU-D.
>
> The star notation refers to the Kleene star, and we use it to refer to sequence space, i.e. $V* = ⋃_{n∈ℕ} V^n$. We overhauled some of the notation in the paper to improve clarity.
>
> **Regarding proofs in the appendix.** We updated the proofs accordingly. Regarding the special case $α=½$, is works in the autoregressive case, which this paper is limited to.

---

> > ### Comment · Reviewer_kNhy · 2022-12-07
> > **Response to authors**
> >
> > I thank the authors for writing the response. I have now looked at what I think is the revised version of the paper (titled 'iclr2023.pdf' in the anonymous github repo). While I am happy with the inclusion of an ablation study, I still have some problems with the presentation of the paper. Overall, I believe that the revised version does not improve much over the original manuscript and will retain the score of 5.
> >
> > __Notations:__
> > I urge the authors to include these explanations of the notations (such as $\psi$ and $*$) in the paper, which I still believe is non-standard. While I understand what they mean now, I don't think it would be the case for the general reader. Likewise, the new notation $[m_t ? x_t : y_t]$ is introduced without sufficient explanation adding to further confusion.
> >
> > __Points not addressed:__
> > Some points from my original review were not fixed in the revised version. In particular,
> > - Appendix B.1 is still incomplete
> > - Plots are not improved. In particular, label sizes should be larger.

---

### Official Review · Reviewer_x2ff · 2022-10-23

**Confidence:** 4
**Correctness:** 3
**Technical Novelty And Significance:** 2
**Empirical Novelty And Significance:** 2
**Recommendation:** 3

**Clarity, Quality, Novelty And Reproducibility:**

## Clarity

The idea is clear but the notations are inconsistent which makes the overall clarity of the paper low. Additional details are also needed regarding the output and update function of the model.

## Quality

The paper lacks key analysis such as expressivity of the resulting function as well as key experments such as a head to head comparison with GRU-D (for instance on synthetic data).

## Novelty

The paper builds upon a lot of existing work and positions itself very close to GRU-D but with the ability to allows for complex eigenvalues in the state update matrix. The idea explored in itself is novel and will eventually provide a solid paper.

## Reproducibility

The paper is currently not reproducible from the manuscript alone. Indeed, some keys components are still required for full reproducibility (such as the output function). The code has been packaged into a python module.

**Details Of Ethics Concerns:**

I have not ethical concern.

**Strength And Weaknesses:**

## Strengths

- The problem tackled in this paper is very important. Namely, how do we build better latent state space models with irregular and missing data.
- The idea of using a linear hidden ODE is nice as it theoretically can express any complex dynamical system, with a sufficiently large hidden process and expressive observation / embedding function.
- The desiderata and objectives for a desireable time series model are clearly stated
- The figure 1 regarding the abstraction for a latent space model is very clear and effective

## Weaknesses
### Notation
An important improvement consits in improving the notation.
- In the problem formulation, it's not clear what $P:\mid t \mid - F$ actually means.
- Introducing a mask instead of "not-missing" would be appreciated too.
- More importantly, the authors seem to mix $z$ and $x$ over the whole manuscript which makes it unecessarily hard to read. For instance, in Section 4, it seems clear that $z$ stands for the latent process while $x$ is the observation. Now, in section 5, $x$ is now used for the latent space ! (And in Algorithm 2, $z$ is used again). $\hat{x}$ is also sometimes used for the latent space (equatiions 9).

This is a very important point that would greatly benefit the readibility of the paper.

### Lemma 2

First, it seems that Lemma 2 lacks a proof. What is more, you mention backward stability in D3. How does the matrix exponential fares in terms of differentiability ? I would assume it's not super stable either ?

Also, if your hidden process is forward stable, what does it say about your whole model ? I guess it depends on the output function that you are choosing. So in a sense, you're not ensuring forward stability of your whole state space model.

### Output function

If I understand correctly, your encoding/update function (from observation space to hidden space) is given by Equation 9. What is missing in your manuscript is a discussion of the output function and how it is linked to the update function.

### Expressivity

One of the important point of that paper is that the desiderata should be achieved without trading for expressivity. In the introduction, the authors hint at the fact that this model can be as expressive as non-linear latent dynamcis models. However, this statement does not get enough focus in the paper. Specifically, I believe an in-depth discussion of the expressivity of the presenteed model should be present in the paper.

### Experiments

- Regarding the non-linear vs linear Kalman cell. It would be nice to have an experiment comparing both approaches.

- The w/hidden version of your model seems to suggest that your latent space and your observation space are the same  (Remark 1)? This seems contradictory with your intro that says that the "observations are nonlinearly mapped into a latent space".

- In general, the experimental section is not very convincing. Indeed, despite the nice layout of the desiderata you need for a good latent state space model, you don't use this blueprint in the experimental section. I believe it would be more impactful to showcase how your model improves upon these metrics.

- It's not clear what "global existence" mean in Table 2.

- Section 6.2 comes a bit out of the blue and lacks context. So it's hard to appreciate its value in the current state of the paper.

### Minor

- end of page 7, I think "git adLinODEnet" is probably a typo.

**Summary Of The Paper:**

This paper proposes a Neural ODE model that implements a filtering approach for time series. In contrast to previous works, this approach relies on mapping the observations on a linear hidden process. The linearity of the underlying ODE allows fast integration and enforcing several desirable properties such as  consistency and forward stability. The authors show improved forecasting performance on MIMIC and weather forecast datasets.

**Summary Of The Review:**

This paper addresses an important problem but currently fails to provide an in-depth investigation of the proposed approach. The experimental section is not convincing and does not reflect the aims of the paper (the 5 desideratas). The notations are inconsistent, making the overall presentation unclear.

---

> ### Author Response · Authors · 2022-11-19
> **Response to Review by x2ff**
>
> **Regarding Notation.** We rewrote the Problem formulation section. One point that needed clarification is the role of the mask. Since the model allows the input of 𝙽𝚊𝙽-values, the typical multiplication with the mask $mₜ⊙xₜ$ does not work, as $𝙽𝚊𝙽⋅0 = 𝙽𝚊𝙽$. We therefore introduce $[mₜ ? xₜ : yₜ]$ as a representation of the 𝚠𝚑𝚎𝚛𝚎-operator.
> We also touched up and make the notation of $z$ and $x$ more consistent. The issue here was that for example in the Kalman-Filter literature, it is more common to use $x$ for the latent state and $y$ for the observations, and we reused that. We updated the notation in the paper to be more consitent.
>
> **Regarding forward stability.** We want to emphasize that this is a property that is only guaranteed at initialization. During training, the model is allowed to learn non-forward stable dynamics. However, this property is crucial, because without it the model tends to blow up during the first epoch of training, but with it, training is stable.
>
> **Regarding the output function.** In this paper, we restrict the evaluation to the forecasting of real-valued autoregressive channels, hence no output function is required. The returned values of Algorithm 1 are the outputs.
>
> **Regarding expressivity.** Koopman theory says that in the limit case of an infinite dimensional latent space, the dynamics can be linearized (cf. “Modern Koopman Theory for Dynamical Systems” by Brunton et al.). We would also reason that the vast improvements over the baselines clearly show that the model is expressive enough to deal with challenging datasets. According to paperswithcode.com, our model is state-of-the-art for these tasks.
>
> **Regarding the experiments.** Upon the suggestion of the reviewer, we ran an ablation experiment using only the linear filter component. The results have been added to Table 3. We also added some forecast plots and figures showing the forward stability in the appendix.
>
> **Regarding statements about hidden units.** In the presented model, hidden variables and latent variables are not the same thing. The hidden variables are part of the data space $𝓧$, and denote virtual channels which are never observed. Internally, the model simply pads the observation matrix with several 𝙽𝚊𝙽-columns.
>
> **Regarding global existence.** By global existence, we mean that the ODE solution exists for all times t. This is not the case for all ODEs, as example 1 in appendix A shows, the ODE $dx/dt = x²$ leads to a finite-time blow up.
>
> **Regarding reproducibility.** We intend to publish both packages `tsdm` and `linodenet` on GitHub and PyPI.

---

### Official Review · Reviewer_M6Ey · 2022-10-24

**Confidence:** 5
**Correctness:** 4
**Technical Novelty And Significance:** 3
**Empirical Novelty And Significance:** 3
**Recommendation:** 6

**Clarity, Quality, Novelty And Reproducibility:**

Clarity:
The full model specification is not in the main text. At least, the distributions of state noise (if any) and measurement(observation) noise are needed.
More elaboration is needed why the nonlinear Kalman cell is needed for a linear model.
It might be a little confusing or inconsistent using $x$ as ODE variable and then as the measurement (observation).

Quality:
The loss or objective function is not in the main text. It is important as there are trainable parameters.
It is unclear how those parameters are trained, in an offline or online fashion.
The speed of proposed method was not shown in the main text even though claimed fast.
It is unclear why the proposed method performed better than other methods. It's possibly that the dynamics in the datasets happen to be (near) linear, the nonlinear dynamics are harder to train, or by other reasons. It would make the work comprehensive to show the performance with/without model mismatch on synthetic data.
For reader's information and formality, more descriptions are needed in the captions of figures.


**Strength And Weaknesses:**

Strength:
The continuous-time dynamics, specified by ODE, allows for irregular sampling and missing values.
The linear ODE specification eases the integrator and computational cost.
The linear specification enables Kalman filter
The approach exhibits forward stability and self-consistency.

Weaknesses:
The linear model limits the power of expressive and rules out such as line attractors, limit cycles, strange attractors and etc.
The manuscript seems finished in a hurry. It lacks information.


**Summary Of The Paper:**

In this work, the authors propose to forecast irregular time series using state-space model whose dynamics is specified by a linear ODE. The proposed method meets several desirable properties to time series forecast such as self-consistency, forward stability and allowing for missing values. The experiments show the proposed method achieved competitive performance.

**Summary Of The Review:**

This work aims to forecast irregular time series using state-space model whose dynamics is specified by a linear ODE. The proposed method meets several desirable properties to time series forecast such as self-consistency, forward stability and allowing for missing values.
Several key information seems missing. The writing can be improved.

---

> ### Author Response · Authors · 2022-11-19
> **Response to Review by M6Ey**
>
>
> **Regarding the dynamical behaviour of LinODEnet.** Note that despite having linear dynamics in the latent space, the model itself is non-linear due to the non-linear encoder/decoder. This is in fact a common paradigm in machine learning: data that is not solvable by a linear model can be solved by a linear model through a non-linear encoding. For instance, most classification models work this way - their last layer is effectively a logistic regression. For dynamical systems, Koopman-Theory shows that indeed a non-linear dynamical system $dx/dt = f(t, x)$ can be expressed as a latent linear dynamical system $dz/dt = K(z)$, where $z=φ(x)$ and $K$ is a linear operator (cf. “Modern Koopman Theory for Dynamical Systems” by Brunton et al.).
>
> Usually, in order to get exact representation, the latent space needs to be infinite dimensional. Since we are only mapping into a finite dimensional latent space, we cannot expect exact representation, but only approximate representation. This is in-line with the usual universal approximation theory of neural networks.
>
> This also explains why we need a non-linear filter, as the model as a whole is non-linear. We added an ablation experiment that shows a significant drop in performance when using only a linear filter.
>
> We improved section 5 by re-adding a short section on the encoder that made the cut in the first submission due to the page limit.
>
> **Regarding notation.** We overhauled and clarified the notation in the paper.

---

> > ### Comment · Reviewer_M6Ey · 2022-12-05
> > **Response to authors**
> >
> > Thank the authors for the responses. I have raised the score.

---

### Official Review · Reviewer_gake · 2022-10-25

**Confidence:** 3
**Clarity, Quality, Novelty And Reproducibility:** The paper is very clear and of good q…
**Correctness:** 3
**Technical Novelty And Significance:** 3
**Empirical Novelty And Significance:** 3
**Recommendation:** 6

**Strength And Weaknesses:**

The theoretical guarantees of the model are nicely derived and clearly presented, and Table 2 is a nice summary of the theoretical advantages of the proposed model.

The empirical evaluation is compelling, but is on synthetic benchmark datasets only. Including results on a real dataset would make the paper stronger. Even without ground truth, including some visualization of the learnt dynamics vs true dynamics and other models for real datasets would be a great addition to the paper (for example, modeling ocean dynamics)

Figure 1 should also be updated. I think it is of poor quality and doesn't give much insight into the model.

Finally, the OBSERVATIONS section (6.2.) is interesting as it shows the Eigen-values of the kernel matrix, and the authors speculate that the learnt dynamics are periodic. There is space to include some true and inferred dynamics to visualize the periodic signal (or maybe include it in the appendix)

**Summary Of The Paper:**

The authors propose a novel Neural ODE model that embeds the observations into a latent space with dynamics governed by a linear ODE.
They carefully show that the model satisfies self-consistency, which allows forecasting irregularly sampled time series and have some numerical stability guarantees. They evaluate the performance on medical and climate synthetic datasets, where the model outperforms the similar state of the art models.

**Summary Of The Review:**

The paper is very clear and shows the the method outperforms previous state of the art models. However, I think the paper would gain in quality with better visualization of the learnt dynamics, in synthetic and real datasets.

---

> ### Author Response · Authors · 2022-11-19
> **Response to Review by gake**
>
> **Regarding benchmark datasets.** Please note that all datasets are real-world datasets. MIMIC-III and MIMIC-IV consist of de-identified health-care data of patients who stayed at the critical care unit of the Beth Israel Deaconess Medical Center. The time-series consists of measurements such as heart-rate, blood-pressure, blood-oxygen, prescriptions, etc. USHCN is a real world climate dataset consisting of measurements from over a thousand different climate stations across the US.
>
> **Regarding Figure 1.** We slightly updated the figure up to increase legibility, but more detailed feedback would be welcome.
>
> **Regarding visualizations.** We added a forecast plot showing the learned trajectories.

---

> > ### Comment · Reviewer_gake · 2022-12-07
> > **Re: Response to Review by gake**
> >
> > Thank you for addressing my comments!

---

### Author Response · Authors · 2022-11-19
**Updated submission**

Unfortunately, we missed the November 18 deadline, due to overlapping work with a pre-planned week-long project meeting. Also, the notification E-Mail could have been clearer about the time-zone of the rebuttal submission. When we tried to upload a revision this morning (within 18. November Anywhere on Earth timezone), it was no longer possible.

# We uploaded the revision of our submission to an anonymous GitHub repository: https://anonymous.4open.science/r/iclr2023-F8CB.

We hope that it still can be considered, given that we and also the reviewers have already invested a lot of time.

---

### Author Response · Authors · 2022-11-19
**New ablation Experiments**

Due to the suggestions of the reviewers, we ran 5 ablation experiments:

1. With a diagonal kernel parametrization.
2. With a linear filter.
3. With a GRU-Filter that is not idempotent (mask values and concatenate mask)
4. With a trivial encoder. (identity instead of a ResNet)

The results can be found in Table 3.

---

### Decision · Program_Chairs · 2023-01-20

**Decision:**

Reject

**Justification For Why Not Higher Score:**

Please see weaknesses listed above.

**Justification For Why Not Lower Score:**

N/A.

**Metareview: Summary, Strengths And Weaknesses:**

Summary: This paper proposes an approach for time-series forecasting based on encoding observations into a latent space where the dynamics is supposedly linear so that costly numerical integration of general Neural ODE's can be replaced by the matrix exponential. This is paired with a Kalman-like filter component. The paper argues that this construction satisfies a desirable self-consistency property, and can handle missing values, together with some numerical stability guarantees.

Strengths: The problem of learning latent space models for irregular/missing-value time series problems is well motivated and relevant for many applications. The proposal of using a linear latent structure is natural and the method appears to be highly competitive relative to baselines.  The desiderata for a desirable time series model are clearly stated.

Weaknesses:   The reviewers did not find the experimental section to be entirely convincing in terms of the desideratas setup earlier. The experiments section feels a little rushed and lacking in a comprehensive analysis, e.g. of whether linearity in latent spaces is too restrictive There were also significant presentation issues highlighted, somewhat amplified by inconsistent notation.